# Effect of Remote Ischaemic Preconditioning on Perioperative Endothelial Dysfunction in Non-Cardiac Surgery: A Randomised Clinical Trial

**DOI:** 10.3390/cells12060911

**Published:** 2023-03-16

**Authors:** Kirsten L. Wahlstrøm, Hannah F. Hansen, Madeline Kvist, Jakob Burcharth, Jens Lykkesfeldt, Ismail Gögenur, Sarah Ekeloef

**Affiliations:** 1Center for Surgical Science, Department of Surgery, Zealand University Hospital, Lykkebækvej 1, 4600 Køge, Denmark; 2Section of Experimental Animal Models, Department of Veterinary and Animal Sciences, Faculty of Health and Medical Sciences, University of Copenhagen, Ridebanevej 9, 1871 Frederiksberg C, Denmark; 3Department of Clinical Medicine, University of Copenhagen, Blegdamsvej 3B, 2200 Copenhagen N, Denmark

**Keywords:** remote ischemic preconditioning, endothelial function, nitric oxide bioavailability, surgical stress, oxidative stress, reactive hyperaemia index, L-arginine, ADMA, biopterin

## Abstract

Endothelial dysfunction result from inflammation and excessive production of reactive oxygen species as part of the surgical stress response. Remote ischemic preconditioning (RIPC) potentially exerts anti-oxidative and anti-inflammatory properties, which might stabilise the endothelial function after non-cardiac surgery. This was a single centre randomised clinical trial including 60 patients undergoing sub-acute laparoscopic cholecystectomy due to acute cholecystitis. Patients were randomised to RIPC or control. The RIPC procedure consisted of four cycles of five minutes of ischaemia and reperfusion of one upper extremity. Endothelial function was assessed as the reactive hyperaemia index (RHI) and circulating biomarkers of nitric oxide (NO) bioavailability (L-arginine, asymmetric dimethylarginine (ADMA), L-arginine/ADMA ratio, tetra- and dihydrobiopterin (BH_4_ and BH_2_), and total plasma biopterin) preoperative, 2–4 h after surgery and 24 h after surgery. RHI did not differ between the groups (*p* = 0.07). Neither did levels of circulating biomarkers of NO bioavailability change in response to RIPC. L-arginine and L-arginine/ADMA ratio was suppressed preoperatively and increased 24 h after surgery (*p* < 0.001). The BH_4_/BH_2_-ratio had a high preoperative level, decreased 2–4 h after surgery and remained low 24 h after surgery (*p* = 0.01). RIPC did not influence endothelial function or markers of NO bioavailability until 24 h after sub-acute laparoscopic cholecystectomy. In response to surgery, markers of NO bioavailability increased, and oxidative stress decreased. These findings support that a minimally invasive removal of the inflamed gallbladder countereffects reduced markers of NO bioavailability and increased oxidative stress caused by acute cholecystitis.

## 1. Introduction

More than 200 million adults undergo major non-cardiac surgery every year and the numbers are increasing [1]. While surgery has the potential to improve life quality and expectancy, surgical procedures also lead to physiological alterations defined as ‘surgical stress’. The alterations affect all organ systems and comprise hypercoagulability, systemic inflammation, immunological disturbances, and endothelial dysfunction [2,3]. 

A healthy endothelium regulates vascular tone, blood fluidity, inflammatory processes, and controls coagulation, while endothelial dysfunction is associated with unfavourable vascular changes, such as loss of endothelium-dependent vasodilation and adoption of a pro-thrombotic endothelial cell phenotype, which are induced by pro-inflammatory processes [4]. Endothelial nitric oxide (NO) is a pluripotent and important signaling molecule acting relaxant in the vasculature. It is produced by the endothelial NO synthase (eNOS) from L-arginine in response to shear stress or autocoids, such as bradykinin and adenosine [5]. The development of endothelial dysfunction is mainly due to a reduced bioavailability of NO, which can be explained by a reduced synthesis and an inactivation of eNOS by oxidative stress [6,7]. There are different ways of assessing endothelial function. Asymmetric dimethyl arginine (ADMA) is an endogenous eNOS inhibitor competing with L-arginine for the binding site, and the NO production is often expressed as the L-arginine/ADMA ratio [6]. A functional method to evaluate the endothelial function is to measure the pulse amplitude in a finger both during reactive hyperaemia and at rest. A reactive hyperaemia index (RHI) is then calculated as the ratio of the two measurements [8]. The digital response to hyperaemia reflects, in part, the NO-dependent vasodilator function of the microcirculation [9,10].

Endothelial dysfunction and its severity are associated with cardiovascular morbidity. Several studies imply that both coronary and peripheral endothelial dysfunction predict disease progression and cardiovascular events [11,12,13,14,15]. More importantly, endothelial dysfunction correlates with myocardial injury and major adverse cardiac events after non-cardiac surgery [16]. Thus, impaired NO bioavailability might play an imperative role in patients developing myocardial injury after non-cardiac surgery [16,17,18,19], a serious complication associated with increased risk of postoperative mortality [20,21,22]. Identifying a method capable of diminishing endothelial dysfunction in response to surgical stress would therefore be invaluable.

Remote ischemic preconditioning (RIPC) is a procedure with short episodes of ischaemia and reperfusion applied to a limb. This non-invasive procedure might possess systemic anti-inflammatory and anti-oxidative properties when introduced prior to non-cardiac surgery [23,24,25,26,27]. Furthermore, RIPC has been shown to reduce myocardial injury in non-cardiac surgery [28]. The mechanisms of RIPC are not fully understood despite decades of experimental and clinical research [29,30]. RIPC activates both a neuronal and a humoral response, as well all exerts systemic effects. In turn these responses activate intracellular signal transduction pathways such as the protein kinase C pathway and the reperfusion injury salvage kinase pathway. The preservation of mitochondrial functions and gene expression modulation are thought to be central mechanisms in the systemic effects of RIPC [31]. We hypothesised that RIPC stabilises the function of the endothelium during surgical stress of non-cardiac surgery. As such, our primary aim was to investigate the effect of RIPC on perioperative endothelial function, including markers of NO bioavailability and oxidative stress, in patients undergoing sub-acute laparoscopic cholecystectomy due to acute cholecystitis. Our secondary aim was to explore changes in endothelial function as a result of subacute laparoscopic cholecystectomies performed due to acute cholecystitis.

## 2. Materials and Methods

### 2.1. Trial Design and Setting 

The trial was a single centre, randomised clinical trial including patients from the Department of Surgery, Zealand University Hospital in Denmark. It was designed to test the effect of RIPC on postoperative endothelial function in patients diagnosed with acute cholecystitis undergoing subacute laparoscopic cholecystectomy. Patients were consecutively screened and included after providing oral and written informed consent between September 2019 and September 2021. The trial was approved by The Danish Data Protection Agency (no. REG-020-2019) and by the Regional Ethics Committee of Region Zealand Denmark (no. SJ-762). The trial was registered at ClinicalTrials.gov (no. NCT04156711) and reported according to the CONSORT Statement [32]. 

### 2.2. Participants

Inclusion criteria were adults (≥18 years) diagnosed with acute cholecystitis and scheduled for subacute laparoscopic cholecystectomy. Patients with symptoms for a maximum of seven days prior to surgery were eligible for inclusion if informed consent could be obtained. Surgeries were performed between 8 A.M. and 8 P.M. Patients fulfilling the following criteria were excluded: surgery within 30 days of inclusion, circumstances preventing RIPC on the upper extremity (e.g., fractures), simultaneous per operative endoscopic retrograde cholangiopancreatography (rendezvous ERCP), synchronous cholangitis, synchronous pancreatitis, or pregnancy. 

Demographic data on all patients, including co-morbidities and daily medications, were collected by research personnel. So were pre- and perioperative data, e.g., vitals, preoperative routine blood samples, perioperative data from the surgeon and anaesthesiologist, and data on the postoperative course. All patients were anaesthetised with propofol and remifentanil. The choice of analgesia was up to the anaesthesiologist in charge. 

### 2.3. Randomisation and Blinding

Patients were randomised into an intervention group or control group. A third party generated the allocation sequence at www.randomization.com (accessed on 8 August 2019) (allocation ratio of 1:1 in blocks of six). A sealed, opaque envelope containing the allocation group was opened by an investigator after patients had given their informed consent to participate. Patients were not blinded to the intervention, but the anaesthesiologist, surgeon and surgical staff were blinded. 

### 2.4. Intervention

All patients received standard care according to local guidelines during their hospital stay, regardless of study allocation (standard regimen of (1) antibiotics: intra venous infusion with metronidazole 500 mg every eight hours and piperacillin/tazobactam 4 g every six hours; (2) analgetic: acetaminophen 1 g every six hours and Ibuprofen 400 mg every eight hours, morphine tablets 10 mg pro necessitate; (3) isotonic saline in case of discomfort during fasting prior to surgery (six hours for food and two hours for thin liquids prior to surgery) and (4) glucose–insulin–potassium intravenous infusion during fasting, according to local guidelines, if patient was diagnosed with diabetes). 

Patients in the RIPC group underwent four cycles of five minutes of ischaemia and five minutes of reperfusion of one upper extremity. The intervention was performed with an electronic tourniquet device (Tourniquet 4500 ECL; VBM Medizintechnik, Sulz am Neckar, Germany) placed on one upper arm with a cuff inflation of 200 mm Hg as a minimum. If a patient had a systolic blood pressure >185 mmHg, tourniquet inflation to a minimum of 15 mmHg above the patient’s systolic blood pressure was required. RIPC was performed less than four hours prior to surgery (knife-to-skin). A biphasic pattern of RIPC-protection has been suggested with an ‘early phase’ activated instantly, lasting about 4 h and peaking within that timespan [33,34]. As such, RIPC was carried out prior to anaesthetic induction as an attempt to align the timing of the RIPC procedure with the peak protective effect. 

### 2.5. Outcomes

Our primary outcome was group-differences in perioperative changes in endothelial function, assessed as the reactive hyperaemia index (RHI), from baseline (preoperative assessment) to 2–4 h and 24 h (postoperative day 1, POD1) after surgery. Our secondary outcomes were group differences in perioperative changes in biomarkers of NO bioavailability: plasma L-arginine, plasma asymmetric dimethylarginine (ADMA), L-arginine/ADMA, plasma tetrahydrobiopterin (BH_4_), plasma dihydrobiopterin (BH_2_) and total plasma biopterin level. NO production was expressed as the ratio between L-arginine and ADMA [6]. The BH_4_/BH_2_-ratio was used as an indirect measure of the level of oxidative stress. In an oxidative stress-free environment BH_2_ levels are close to undetectable and all measurable biopterin are in the reduced form of BH_4_. However, under conditions of oxidative stress BH_4_ is highly susceptible to oxidation resulting in the formation of BH_2_. As such, BH_4_ decreases significantly and BH_2_ increases, correlating a lower BH_4_/BH_2_ ratio to more oxidative stress [35]. 

### 2.6. Data Sources 

Non-invasive digital pulse amplitude tonometry (Endopat2000; Itamar Medical Ltd., Caesarea, Israel, Software Version 3.7.x) was used to assess endothelial function at patient inclusion (preoperative), 2–4 h after surgery and 24 h after surgery [36]. While the patient was in a supine position, a finger probe was placed on each index finger. The Endopat2000 assesses the digital pulse amplitude during five minutes of rest, during five minutes of blood flow occlusion (with a blood pressure cuff inflated to a supra-systolic pressure on the upper extremity), and finally during the subsequent five minutes of hyperaemia as the cuff is deflated [8]. A reactive hyperaemia index (RHI), a measure of endothelial function, is calculated automatically by the EndoPat2000 system. It is a ratio between the digital pulse amplitude during hyperaemia and at rest. The pulse amplitude is measured at both fingers in all three phases, allowing one finger to serve as a control, adjusting for systemic effects. Furthermore, as the RHI is analysed automatically any interobserver variability is diminished [8]. The proposed cut-off values for a normal RHI are ≥2.10, whereas <1.67 is considered abnormal and the range in between as borderline [37].

Whole blood was withdrawn into ethylenediamine tetra-acetic acid (EDTA) tubes. Blood was collected just prior to the Endopat assessment during patient inclusion (preoperative), 2–4 h after surgery and 24 h after surgery. The initial sampled blood was disposed.

For analysis of biopterin, blood was collected in EDTA tubes (3 mL) with 75 μL freshly made 1,4-Dithioerythritol (DTE) solution (50 mg DTE in 10 mL Milli-Q water) added to minimise oxidation during analysis [38]. Samples were mixed gently before centrifugation (2000× *g*, 10 min, 4 °C) and plasma was aliquoted and stored immediately at −80 °C in Eppendorf tubes until analysis. Quantifying biopterin concentrations was determined using high-performance liquid chromatography (HPLC) with fluorescence detection employing iodine oxidation, as previously described [39].

For L-arginine and ADMA analysis, EDTA tubes (6 mL) containing whole blood were centrifuged at 2000× *g*, for 10 min, at 4 °C and plasma was aliquoted and stored immediately at −80 °C. HPLC with fluorescence detection was used for quantification [40].

### 2.7. Statistical Analyses

This was an explorative study and as such, no sample size calculation was performed. Patients undergoing RIPC, as described in the protocol, and having one preoperative blood sample and at least one postoperative blood sample withdrawn were included for analyses in this study. 

Categorical data are expressed as units (n, %) and compared between control patients and patients undergoing RIPC by Pearson’s Chi-squared test or Fisher’s Exact Test. Continuous data were visualised by histograms and quantile–quantile plots of residuals to control the data distribution and equality of variances. Parametric data were compared by Student’s *t*-test and presented as mean and standard deviation (SD), while non-parametric data were compared by the Mann–Whitney U test and presented as medians and ranges. Benjamini and Hochberg’s method to decrease the false discovery rate was applied to significant results as an adjustment for multiple testing. 

To analyse changes in outcomes over time with repeated measurements, we applied a constrained linear mixed model (using the ‘LMMstar’ package (version 0.7.6) in R [41]) including follow-up time (categorical) as a fixed effect. To account for the correlation in the repeated measurements and possible variance heterogeneity over time, we assumed an unstructured covariance pattern. We adjusted for duration of surgery and blood loss between groups, due to numerical differences in the variables between groups. This described method was applied for group differences according to the primary and secondary outcomes. The constrained linear mixed model for randomised studies uses one population mean at baseline, assuming that as all random samples are drawn from the same population, they must share the same true population mean. A linear mixed model was applied to a merged population of controls and patients undergoing RIPC when analysing the effect of surgery on RHI and concentrations of NO bioavailability markers (L-arginine, ADMA, L-arginine/ADMA, BH_4_, BH_2_, and total plasma biopterin). The population was merged to avoid reduction in sample size. This approach was accepted after primary analyses revealed neutral findings on the effect of RIPC. Furthermore, subgroup analysis on the effect of surgery in controls only showed no difference in significance-level of the results. Analyses were considered significant at a *p*-value of <0.05.

We did a post-hoc power analysis applying the observed group difference and variance in the measurements of our primary outcome, RHI. Based on these estimates, our study had a power of 0.53, and to reach a power of 0.80, a sample size of 57 patients in each group would be necessary. 

All statistical analyses were performed using RStudio (RStudio Team (2019) [42]). 

## 3. Results

### 3.1. Patients

Sixty patients undergoing surgery due to acute cholecystitis were included in this study during a 24-month period. The details on the patient flow are illustrated in Figure 1. A total of 210 patients admitted with acute cholecystitis met the inclusion criteria. However, 132 patients either declined to participate, relocated from other hospitals, or had surgery shortly after diagnosis, which was incompatible with inclusion and intervention. Hence, 78 patients were enrolled in the study and 18 were excluded immediately after surgery as perioperative endoscopic retrograde cholangiopancreatography was performed due to choledocholithiasis (an exclusion criteria). A such, 60 patients randomly allocated to either a RIPC group (n = 30) or a control group (n = 30) were analysed.

The demographics of the study population, including distribution of comorbidities, daily medication, and ASA group, were comparable between the control group and intervention group. Details are shown in Table 1. All patients underwent general anaesthesia induced and maintained with propofol and remifentanil. Pre-medication and surgical characteristics did not differ between groups. Eleven patients had a surgical drain inserted perioperatively due to either gall bladder perforation or bleeding. After discharge, eight patients were readmitted, three of whom had complications related to their surgery (intraabdominal abscess, n = 2) or gallstone disease (cholangitis, n = 1). The remaining five patients had abdominal pain with spontaneous relief. 

RIPC was applied and completed within 4 h to skin incision, with a mean time from completed RIPC-procedure to skin incision of 2 h and 20 min (range 40–240 min).

### 3.2. The Effect of RIPC on Endothelial Function and Nitric Oxide Bioavailability

All 60 patients had a preoperative measurement of RHI. All but one patient had at least one follow-up measurement within 24 h after surgery. The reason for missing measurements of RHI in one patient was postoperative nausea (four hours after surgery) and early discharge (<24 h after surgery). Patients in our study had a preoperative RHI value of 1.82 (95% CI 1.70–1.93). There were no differences in RHI between patients in the RIPC and control group over time (from preoperative to 24 h after surgery) (*p* = 0.07, Figure 2). 

Fifty-four patients had blood samples withdrawn for analyses at all three timepoints. One patient had blood samples withdrawn preoperatively and 2–4 h after surgery, whereas five patients had samples drawn preoperatively and 24 h after surgery. There were no differences in concentrations of L-arginine (*p* = 0.36), ADMA (*p* = 0.72), L-arginine/ADMA-ratio (*p* = 0.69), BH_4_ (*p* = 0.07), BH_2_ (*p* = 0.38), BH_4_/BH_2_-ratio (*p* = 0.11), or total biopterin concentration (*p* = 0.22) over time between patients undergoing RIPC or patients in the control group, Figure 2.

### 3.3. The Effect of Surgery on Endothelial Function and Nitric Oxide Bioavailability

RHI did not change significantly in response to surgery (*p* = 0.83, Figure 3). However, both L-arginine and L-arginine/ADMA increased as an overall response to surgery (*p* < 0.001 and *p* = 0.01, respectively, Figure 3). L-arginine concentration preoperative was 44.8 (39.9–49.7) μmol/L and increased with +15.2 (7.55–22.8) μmol/L (*p* < 0.001). The preoperative L-arginine/ADMA ratio was 34.2 (28.1–40.4) and increased with +13.3 (2.83–23.7) 24 h after surgery (*p* = 0.01). The ratio had a numerical but non-significant decrease from the preoperative level till 2–4 h postoperatively −5.44 (−14.7–3.82). However, from the ratio at 2–4 h postoperatively until POD1, a significant increase was observed (*p* = 0.003). The overall effect of surgery on BH_4_/BH_2_ ratio was a reduction (*p* = 0.01). The preoperative BH_4_/BH_2_-ratio was 4.62 (1.71–7.52) and decreased significantly at 2–4 h after surgery with −2.28 (−4.35–−0.20), *p* = 0.04. This decrease was also present and significant 24 h after surgery (−3.13 (−6.06–−0.19)), *p* = 0.03, Figure 3. There were no significant changes in relation to surgery in concentrations of ADMA, BH_4_, BH_2_, or total biopterin (Figure 3).

## 4. Discussion

We did not demonstrate any effect of RIPC on endothelial function, assessed as a reactive hyperaemia index, 2–4 h or 24 h after laparoscopic cholecystectomy compared with the preoperative assessment. Neither did RIPC affect markers of NO bioavailability 2–4 h or 24 h after surgery compared with the preoperative measurements of these biomarkers. Surgery alone, however, did impact on plasma L-Arginine/ADMA ratio and BH_4_/BH_2_ ratio.

The effect of RIPC on RHI has been investigated in one other trial with patients undergoing subacute hip fracture surgery, where endothelial function was assessed as a point-assessment of RHI on postoperative day one. The study found no significant effect of RIPC on postoperative RHI [43]. Apart from the RIPC procedure being initialised in the operating room just prior to surgery, the definition, duration, and number of cycles were identical to our RIPC intervention [43]. While no other studies, to our knowledge, have investigated the effect of RIPC on postoperative endothelial function in patients undergoing non-cardiac surgery, it has been investigated in patients with acute myocardial infarction prior to percutaneous coronary intervention with a significant effect lasting until one week after the intervention [44]. Studies in healthy volunteers [45,46] and in hypertensive patients [47] have also shown a beneficial effect of RIPC on endothelial dysfunction induced by ischaemia-reperfusion injury to a limb. However, the method of assessing endothelial function was different from ours, as either invasive flow-mediated dilation [45,47] or calculation of vascular conductance [46] was used. Furthermore, essential differences in the pathophysiology of endothelial dysfunction induced by ischaemia-reperfusion and that of a systemic surgical stress response must be expected. Moreover, our patients had several days of symptomatic cholecystitis prior to application of RIPC. As such, they were not only exposed to a surgical stress response but had potentially suffered from additional ischaemic gall bladder tissue in the days preceding RIPC and surgery. This could affect the beneficial effect of RIPC. A study design assessing the effect of RIPC on endothelial function in patients undergoing, e.g., elective laparoscopic cholecystectomy could clarify this question.

Another way of investigating the endothelial condition is to measure the endothelial NO bioavailability, as reduced NO bioavailability is essential in the pathophysiology of endothelial dysfunction [48]. Endothelial NO is primarily produced by endothelial nitric oxide synthase (eNOS) from L-arginine. As ADMA is an endogenous eNOS inhibitor competing with L-arginine for the binding site the NO production is often expressed as the L-arginine/ADMA ratio. ADMA and NO bioavailability have been proven to correlate with a wide range of cardiovascular pathologies, e.g., myocardial infarctions, hemodynamic instabilities, acute heart failure and peripheral arterial disease [49,50]. Moreover, an essential cofactor for NO synthase is BH_4_. It modifies the enzyme’s activity in coexistence with its reduced form, BH_2_. They compete for the enzyme’s binding site, but when BH_2_ is bound to eNOS, it causes an uncoupling of the synthase in contrast to BH_4_. Per se, the BH_4_/BH_2_ (or total biopterin) ratio has been claimed to be a main parameter in defining the extent of eNOS uncoupling [48,49]. Uncoupling shifts the production of NO to superoxide and peroxynitrite, initiating a cascade of increased oxidative stress, and aggravating the reduced NO bioavailability [51,52].

To our knowledge, no other study has repeatedly investigated the effect of RIPC on circulating levels of L-arginine, ADMA or biopterin in patients undergoing non-cardiac surgery. However, one study reported significant protection of RIPC in contrast-induced acute kidney injury with simultaneously decreased levels of ADMA. Furthermore, RIPC has been demonstrated to reduce ADMA levels after induced ischaemia-reperfusion in healthy volunteers [53]. A beneficial effect of RIPC on other biomarkers of oxidative stress, such as malondialdehyde and superoxide dismutase, has been reported in both minor and major non-cardiac surgery [23,24,25,26]. 

We did not demonstrate an effect of surgery on RHI. There are several studies in elective minor and major non-cardiac surgery reporting a reduction of RHI in response to surgical stress [54,55,56]. An explanation for this difference might be that patients in our study suffered from acute inflammation for several days and had a decreased endothelial function already at preoperative measurement, as inflammation is known to affect endothelial function [57]. In accordance with the proposed cut-off values for RHI, the preoperative value in our study was in the range of ‘borderline’ between normal and abnormal values [37]. Another study exploring the endothelial function after major emergency abdominal surgery described RHI to be suppressed after surgery, but no preoperative value was measured, so the endothelial dysfunction could have been present prior to the acute surgical intervention, as in our study [16]. 

We found L-arginine and L-Arginine/ADMA ratio to be suppressed preoperatively but both increased significantly 24 h after surgery. These findings are consistent with previous published surgical studies [16,54,55]. Our results suggest that NO bioavailability is impaired because of the patients’ acute disease and although surgery causes a systemic stress response also capable of causing endothelial dysfunction [16,54,55], the minimally invasive laparoscopic removal of the gallbladder countereffects the systemic endothelial dysfunction caused by acute cholecystitis. This is in line with the clinical situation as patients with acute cholecystitis are often bedridden before surgery but ready for discharge within the same day as surgery is effectuated. Moreover, we found that the BH_4_/BH_2_-ratio changes significantly in the perioperative period. Compared to previous studies in non-cardiac surgery, it had a high preoperative level [16,54,55] and decreased significantly up to 24 h after surgery (*p* = 0.01). This supports our findings of increased NO biomarker availability after surgery. 

Our trial has strengths and limitations. It is one of the first studies to address the effect of RIPC on RHI in non-cardiac surgery and it is the first to examine the effect of RIPC on circulating levels of L-arginine, ADMA, and biopterin. Moreover, it was a randomised clinical trial and we excluded patients undergoing rendezvous ERCP or having synchronous cholangitis or pancreatitis to avoid introducing heterogeneity of the surgical or systemic stress response in our patient population.

Attempts were made to include patients consecutively, but this was challenged due to a period of national COVID-19 lockdown and the logistic challenges of always having research staff available at the hospital. The intervention took 40 min and a maximum of 4 h from RIPC-procedure until surgery was accepted, but occasionally operating beds were re-prioritised an patients included in the study were postponed, resulting in exclusion due to our defined intervention-to-surgery time threshold. The trial was exploratory, and no sample size calculation was performed. It has been demonstrated that RHI measurements and data vary depending on the type of surgery and the population investigated [16,43,54,55]. Nonetheless, the lack of sample size calculation introduces the risk of type II errors. A post hoc power analysis based on our RHI measurements revealed a statistical power of 0.53 and a sample size of 57 patients in each group would have been necessary to reach a power of 0.8. Furthermore, it could be argued that a 24 h follow-up is too short. The RIPC procedure was applied in patients prior to anaesthesia. Therefore, patients might unknowingly have influenced our results. Though patients were instructed not to talk or move during the RIPC procedure, uneasiness causing movements, fear of discomfort or excitement might have increased blood pressure and attenuated the effect of RIPC. However, studies in awake volunteers have demonstrated effects of RIPC attenuating endothelial ischaemia-reperfusion injury [45,46], nonetheless it is a possible limitation to the study. Our patients underwent general anaesthesia with propofol and it is still debated to what extent propofol inhibits the effect of RIPC. An RCT demonstrated that RIPC abolished cardio-protection (measured by hs-troponin release) in patients undergoing cardiac surgery [58]. On the other hand, a study investigated the interference of anaesthesia on the cardio-protective effect of RIPC and demonstrated that both propofol and sevoflurane blocked the effect of RIPC [59]. Furthermore, in a recent non-cardiac multicentre trial of patients undergoing hip fracture surgery, a multivariable logistic regression showed no interaction between RIPC and type of anaesthesia [28].

In conclusion, RIPC did not influence endothelial function, markers of NO bioavailability, or the biopterin redox state up to 24 h after laparoscopic surgery for acute cholecystitis. In response to acute laparoscopic cholecystectomy, markers of NO bioavailability increased, and the stress-induced redox imbalance reflecting eNOS uncoupling decreased 24 h after surgery. This might reflect that operative removal of the diseased gallbladder counteracts the systemic endothelial dysfunction caused by acute cholecystitis, despite the introduction of minor surgical stress.

Based on our findings, further trials investigating the effect of RIPC on postoperative endothelial function in non-cardiac surgery is needed. Preferably RCTs with both functional tests and biomarker outcomes reflecting endothelial function in patients undergoing elective non-cardiac surgery. An adequate sample size for both primary outcomes and for reasonable subgroup analyses, e.g., type of anaesthesia, is recommended. Moreover, measurements should be repeated for a longer period, e.g., up to 3–4 days, as the surgical stress response can last for several days.

## Figures and Tables

**Figure 1 cells-12-00911-f001:**
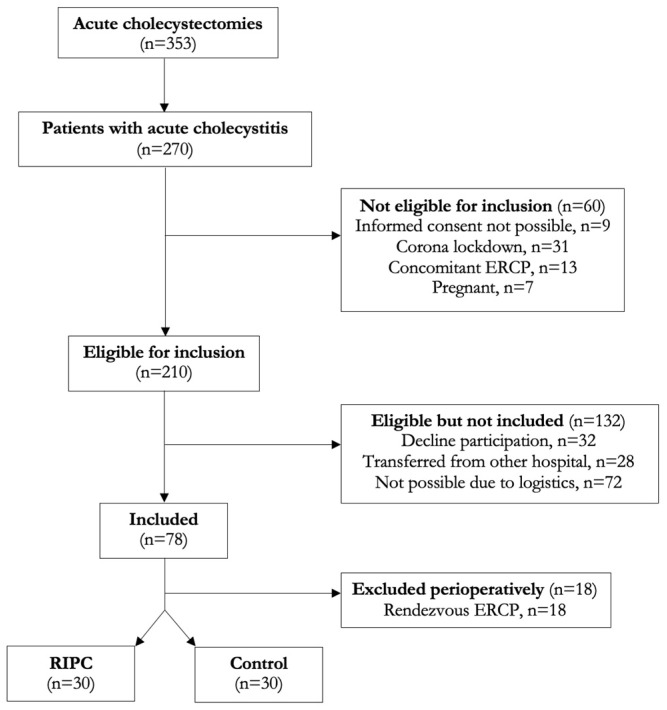
Patient selection flow chart.

**Figure 2 cells-12-00911-f002:**
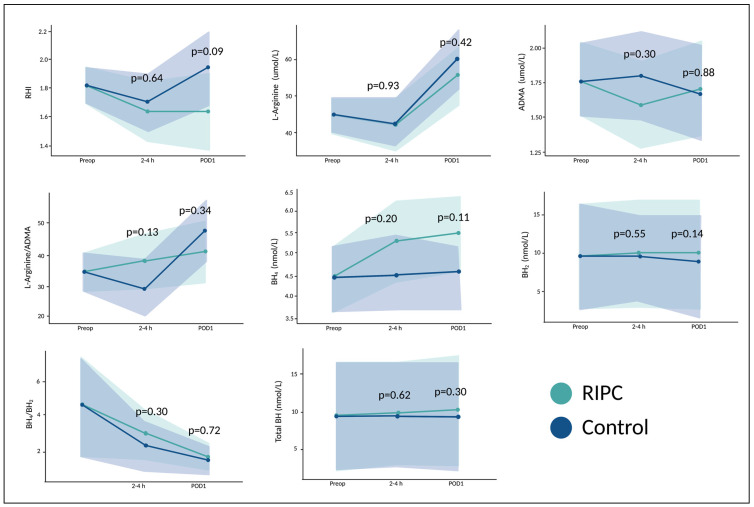
Endothelial function and markers of NO bioavailability stratified by remote ischemic preconditioning in patients undergoing surgery. Means and 95% CI of reactive hyperaemia index (RHI), L-arginine, asymmetric dimethylarginine (ADMA), L-arginine/ADMA-ratio, tetrahydrobiopterin (BH_4_), dihydrobiopterin (BH_2_), BH_4_/BH_2_-ratio, and total biopterin concentration in relation to surgery in patients undergoing remote ischemic preconditioning (RIPC) and controls. Preop = preoperative; 2–4 h = two–four hours after surgery; POD1 = 24 h after surgery. *p*-values are given for the overall difference between groups. Created with R studio and www.BioRender.com (accessed on 20 February 2023).

**Figure 3 cells-12-00911-f003:**
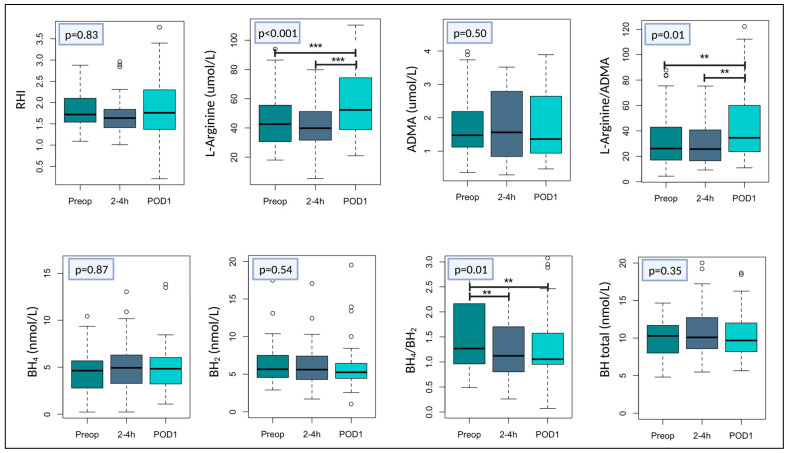
Changes of endothelial function and markers of NO bioavailability in patients undergoing surgery. Boxplots illustrating changes of reactive hyperaemia index (RHI) and plasma concentrations of L-arginine, asymmetric dimethylarginine (ADMA), L-arginine/ADMA-ratio, tetrahydrobiopterin (BH_4_), dihydrobiopterin (BH_2_), BH_4_/BH_2_-ratio, and total biopterin in relation to surgery. Box plots show median with variance with lower and upper hinges representing the 25th and 75th percentile, respectively. White circles represent measurements below or above the 25th and 75th percentile. Preop = preoperative; 2–4 h = two–four hours after surgery; POD1 = 24 h after surgery. *p*-values are given for the overall changes in time. *** *p* < 0.001 ** *p* < 0.05. Created with R studio and www.BioRender.com (accessed on 20 February 2023).

**Table 1 cells-12-00911-t001:** Baseline and peri-operative characteristics of patients stratified by remote ischaemic preconditioning (RIPC) and controls receiving standard care.

Baseline Characteristics	RIPC (n = 30)	Control (n = 30)	*p*-Value
Sex, No. (%)			
Males	20 (67)	17 (57)	0.43
Age, Median (range); years	54 (29–80)	49 (33–77)	0.24
Body Mass Index, median (range); kg/m^2^	29.7 (19.7–41.5)	29.3 (19.6–48.0)	0.89
Daily smoking, No. (%)	8 (27)	7 (23)	0.77
Alcohol abuse, No. (%)	1 (3)	2 (7)	0.55
**Comorbidity, ^A^ No. (%)**			
Hypertension	8 (27)	5 (17)	0.35
Hypercholesterolemia	2 (7)	3 (10)	0.64
Diabetes mellitus	2 (7)	3 (10)	0.64
Ischemic heart disease	1 (3)	0 (0)	0.97
Atrial fibrillation	2 (7)	0 (0)	0.49
Previous Stroke	2 (7)	2 (7)	1.0
**Daily Medicine intake, No. (%)**			
Beta-blocker	1 (3)	0 (0)	0.97
Calcium antagonist	4 (13)	1 (3)	0.16
ACE-1/ARB	3 (10)	5 (17)	0.45
NSAID	4 (13)	4 (13)	1.0
Glucocorticoids	0 (0)	2 (7)	0.49
Opioid	3 (10)	4 (13)	0.69
**ASA classification, No. (%)**			0.40
ASA I	11 (18)	10 (17)	
ASA II	14 (23)	18 (30)	
ASA III	5 (8)	2 (3)	
**Preoperative medication, No. (%)**			
Antibiotics	21 (70)	21 (70)	1.0
Opioid	3 (10)	4 (13)	0.69
NSAID	4 (13)	4 (13)	1.0
**Symptomatic days**			
Prior to surgery, median (range)	3 (1–5)	3 (1–5)	0.82
**Surgery**			
Duration, min, median (range)	96 (48–274)	110 (44–240)	0.82
Blood loss, mL, median (range)	150 (20–1200)	250 (20–1550)	0.18
Bile leakage, No. (%)	18 (60)	13 (43)	0.20
Propofol, mg, median (range)	985 (220–1400)	1022 (399–1550)	0.77
**Analgesia 24 h post-operative**			
Acetaminophen			
No. (%)	12 (40)	8 (27)	0.27
mg, median (range)	1 (1.0–2.0)	1 (1.0–3.0)	
NSAID			
No. (%)	5 (17)	5 (17)	1.0
mg, median (range)	1000 (400–1600)	800 (200–1200)	0.47
Morphine			
No. (%)	2 (3)	6 (20)	0.13
mg, median (range)	7.5 (5.0–10.0)	15.0 (5.0–45.0)	0.08
**Clavien–Dindo Classification, No. (%)**			0.80
1	1 (3)	1 (3)	
2	7 (23)	5 (17)	
3a	2 (7)	0 (0)	

## Data Availability

The data presented in this study are available on request from the corresponding author.

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
