# Peer review of "Effect of Remote Ischaemic Preconditioning on Perioperative Endothelial Dysfunction in Non-Cardiac Surgery: A Randomised Clinical Trial"

_cells, 2023, doi:10.3390/cells12060911_

Round 1

Reviewer 1 Report

Non-pharmacological strategies to protect the vasculature from the cellular stress and damage induced by invasive surgical procedures are needed. RIPC has great potential to alleviate such stress and preserve endothelial function in these scenarios. The present study investigated whether RIPC can favorably modify peripheral vascular endothelial function in patients undergoing acute laparoscopic cholecystectomy and influence surrogate biomarkers of NO bioavailability compared with a control standard of care. While the concept is sound and clinically important, there are numerous concerns related to study design, analysis, data presentation, and conclusions that undermine the study. 

1) It is not clear where the tourniquet was placed. Typically, occlusion is on the upper arm, but the authors indicate the forearm. If that is the case, then it is not surprising to see neutral effects with RIPC. The authors should elaborate in more detail on the placement of the occlusion device. 

2) in line with this, an occlusion pressure of 200 mmHg is likely to be insufficient given the comorbidities present and hypertension. Because there was no assessment of deoxygenation or perfusion in the occluded limb, it is difficult to know whether there was full ischemia, which of course will undermine the efficacy of RIPC. To address this issue, it is advised to include the specific data on the combordities - blood pressure, cholesterol levels etc. The average pressure of occlusion should also be reported.

3) There needs to be more detail on the standard of care for the controls. 

4) A primary issue is that lack of power and sample size calculation for a randomized control trial. Without that, which is standard rigor of trials, that the authors say their study aligned with CONSORT, reduces the enthusiasm for and the meaningfulness of the study. In addition, indicating that the study was explorative suggests that the study was not well-conceived, particularly because the authors are introducing uncertainty in the overall design. 

5) The range and mean time after the completion of RIPC to surgery should be reported. 

6) Because the time of surgery was longer in the control group and the level of blood loss was worse, this could be a key factor underlining the neutral findings. These data would need to be included as covariates in the model. 

7) Table 1 should have p values showing the group comparisons. 

8) Line 212 should read '...randomly assigned...' instead of divided

9)Some data appear to be log transformed, yet that is not reported in the results. 

10) the use of propofol and to some extent morphine should be included in the model because both are preconditioning agents and can mask RIPC.

11) The data in fig 2 is not presented well - too small, overlapping error bars and does not show individual values. Consider showing box plots. 

12) the data shown in figure 3 appears to be the combined sample of controls and those receiving RIPC. It is not clear why the authors have chosen to combine the sample. These data will create misleading information and should be removed. Furthermore, there are many subjects that could be classified as outliers - for example the subject with the very high value of L-arg is most likely driving the significance those respective graphs (L-arg/ADMA).

13) The researchers did not directly measure NO bioavailability, only biomarkers that suggest it. Therefore, language indicating that NO bioavailability per se was changed should be changed. 

14) the conclusions in the discussion that surgery alone impacted these data are thus not correct given the sample was combined. All the text in the discussion (lines 334-359) should be omitted. 

15) Discussion lines 394-398 is not correct. There is no indication in the data provided that RHI increased up to 24 hours after surgery with RIPC. It appears the authors have misinterpreted that data because it the control group's value that seems higher at 24 hrs, not the RIPC group. This needs to be omitted and corrected. 

Reviewer 2 Report

This is a well conducted and presented study on the effect of RIPC and cholecystectomy on endothelial function in patients suffering from acute cholecystitis. RIPC has been studied extensively in cardiac surgery but less so in other types of surgery and this study adds to the currently limited evidence, particularly regarding the effect of RIPC on postoperative endothelial function.

Abstract

The fact that the surgery was emergency/urgent cholecystectomy in patients with acute cholecystitis needs to be mentioned early and made clear in the abstract.

The potential of RIPC to exert anti-oxidative and ant-inflammatory properties is not a definite fact and should not be stated as such. The same for the Introduction part.

Introduction

Please briefly describe the definition/pathophysiology of endothelial dysfunction i.e. what does it mean for vasomotor function, prothrombotic state etc.?

A brief explanation of the proposed mechanisms of action of RIPC would be useful.

Material and methods

Please explain the rationale for doing the intervention with the patients awake rather than after induction of anaesthesia.

Data sources: Please include normal values for RHI

Discussion

The fact the RIPC is applied to patients that are having acute cholecystitis and have therefore already suffered a potentially ischaemic event may counteract the beneficial effect of RIPC and that should be part of the discussion. Is it possible that RIPC would have had a beneficial effect if the surgery was elective?

It has been suggested that the use of propofol for anaesthesia may also inhibit the beneficial effect of RIPC and the choice of anaesthesia in the study should also be discussed.

Is it possible that performing the intervention to awake patients might have influenced your results? Please discuss

Strengths of the study: please mention that it is one of the first studies to address the effect of RIPC on RHI in non-cardiac surgery and the first study to examine the effect of RIPC on levels of L-arginine, ADMA and biopectrin as well as the reproducibility of digital pulse tonometry here. The rest of the comments in that paragraph have already been described in the methods and there is no need to be repeated here.

Study limitations: The details regarding sample size calculation and power of the study have already been described in the statistical methods and there is no need to be repeated here.

What further research is needed regarding RIPC and postoperative endothelial function, based on the outcomes of your study?

Other comments

Please be consistent in using either BH2/BH4 or BH4/BH2 throughout the manuscript.

Round 2

Reviewer 1 Report

Thank you for addressing the comments. The manuscript is much improved. 

Reviewer 2 Report

Thank you for all the changes, I think it's much improved.